# Cell Sources for Retinal Regeneration: Implication for Data Translation in Biomedicine of the Eye

**DOI:** 10.3390/cells11233755

**Published:** 2022-11-24

**Authors:** Eleonora N. Grigoryan

**Affiliations:** Koltzov Institute of Developmental Biology, Russian Academy of Sciences, 119334 Moscow, Russia; leonore@mail.ru

**Keywords:** retinal degenerative diseases, retinal regeneration, intrinsic cell sources, regulatory network, ophthalmotherapy

## Abstract

The main degenerative diseases of the retina include macular degeneration, proliferative vitreoretinopathy, retinitis pigmentosa, and glaucoma. Novel approaches for treating retinal diseases are based on cell replacement therapy using a variety of exogenous stem cells. An alternative and complementary approach is the potential use of retinal regeneration cell sources (RRCSs) containing retinal pigment epithelium, ciliary body, Müller glia, and retinal ciliary region. RRCSs in lower vertebrates in vivo and in mammals mostly in vitro are able to proliferate and exhibit gene expression and epigenetic characteristics typical for neural/retinal cell progenitors. Here, we review research on the factors controlling the RRCSs’ properties, such as the cell microenvironment, growth factors, cytokines, hormones, etc., that determine the regenerative responses and alterations underlying the RRCS-associated pathologies. We also discuss how the current data on molecular features and regulatory mechanisms of RRCSs could be translated in retinal biomedicine with a special focus on (1) attempts to obtain retinal neurons de novo both in vivo and in vitro to replace damaged retinal cells; and (2) investigations of the key molecular networks stimulating regenerative responses and preventing RRCS-related pathologies.

## 1. Introduction

Degenerative changes in the animal and human retina related with age or diseases cause visual dysfunction and blindness. The major, well-known retinal disorders are age-related macular degeneration (AMD), retinitis pigmentosa (RP), and glaucoma. This list also includes diabetic retinopathy, proliferative vitreoretinopathy (PVR), Stargardt disease, etc. One opinion is that retinal degeneration, which leads to decreased vision and blindness, is a health problem comparable in prevalence and significance to Alzheimer’s disease and cancer [1,2].

The long-term search for potential RRCSs and comprehensive investigations have led to the discovery of such sources in the eye tissues and to understanding of their biology. There has been progress in the study of genetic intrinsic features of RRCSs and external regulation mechanisms responsible for retention or acquisition of the low-differentiated state of RRCSs, conversion and neural/retinal differentiation. Despite significant differences in the degree of manifestation of RRCSs’ regenerative potencies in vertebrates in the evolutionary series, all of them, one way or another, belong to the same spectrum (Figure 1). The latter comprises undifferentiated or poorly differentiated precursors localized in the ciliary region of the retina, retinal pigmented epithelium (RPE), ciliary body (CB) of the eye, and also Müller glial cells (MGCs). Note that these cell populations are beyond the set of neural cell phenotypes of the retina, but they are mandatory and integrated in the system of its functions. All of these RRCS types that can potentially be an intrinsic regenerative reserve in the retina are considered in numerous reviews summarizing the results of long-term experimental research conducted by several laboratories worldwide [3,4,5,6,7,8,9,10,11,12]. In the present study, efforts were focused on finding ways for practical application of the accumulated knowledge about RRCSs and approaches to regulating their properties and behavior for the purpose of translation to biomedicine of the eye. In this review, an attempt is made to understand the implications for the development of this research trend.

Currently, embryonic progenitors and embryonic, neural, mesenchymal, and induced pluripotent stem cells are considered to be sources of RPE and neural retina (NR) regeneration, as an alternative to autologous mammalian and human RRCSs. Their use provides a wide range of opportunities, while nevertheless posing numerous risks, as is widely discussed in the modern literature [12,13,14,15,16,17]. Thus, an approach based on transplantation of stem cell-derived RPE cells for the treatment of AMD is currently being developed, with clinical trials conducted [18,19,20,21,22]. Being extensively analyzed and discussed in the literature, this topic is beyond the scope of the present review, which mainly provides information about intrinsic RRCSs and evaluates the feasibility of using this information in biomedicine.

## 2. The Main Structure of the Retina

The vertebrate retina is organized by a single plan, which, however, has specific morphological and functional features [22,23]. The retina is a neural, highly structured, stratified tissue where different types of neurons have strict localization, maintaining a stereotypic pattern of the NR (Figure 2). Functionally, it is a sensory tissue consisting of ordered layers whose cells interact with one another and neurons of other layers to provide light perception, receiving, and transmitting visual information. The retina is formed by six major NR cell types and RPE cells. NR cell populations are represented by photoreceptors (rods and cones), bipolars, and horizontal, amacrine, and ganglion cells. Müller glial cells (MGCs), microglia cells, astrocytes, and oligodendrocytes are integrated into the NR. The NR consists of three nuclear layers and two reticulate (plexiform) layers formed by fibers and synaptic contacts of neurons. The outer nuclear layer (ONL) is composed of bodies of photoreceptors, rods and cones. Their processes establish topological and functional connections with RPE. The inner nuclear layer (INL) contains bipolar, horizontal, and amacrine cells. INL interneurons are responsible for the visual signal transmission from photoreceptors to ganglion cells constituting the ganglion layer. The long processes of the ganglion cells compose the optic nerve that transmits information to the visual analyzer of the brain. The outer plexiform layer (OPL) is formed by fibers and synaptic contacts between photoreceptors and bipolar; the inner plexiform layer (IPL), by processes and synapses between bipolar and ganglion cells, as well as by horizontal processes forming connections between horizontal and amacrine cells. Bodies of the MGCs are located in the INL, extending their processes to the inner and outer limiting membranes of the NR, and are involved in their formation. In mammals and human the NR has two vascular supplies, the choroidal vasculature underlying RPE and the vessels of the inner retina (Figure 2). The blood supply to the inner retina is via the central retinal artery, whose branches radiate from the optic nerve head onto the inner retinal surface and then give rise to branches that penetrate into the retina through the INL, IPL, and OPL [24].

## 3. Brief Characteristics of the Major Degenerative Disorders of the Retina

As mentioned above, the general types of retinal degeneration include AMD, glaucoma, RP, and PVR. All of these disorders, except glaucoma, are caused by the loss of cells and cell–cell interactions in the functional light perception system, RPE and NR (Figure 2). AMD, affecting, according to approximate estimates, a quarter or more of the global population aged 65+, is accompanied by the loss of photoreceptors in the maculae region, where the light rays are focused on the retina. There are two forms of AMD: the “dry” (prevailing) and “wet” AMD [25]. With the dry AMD in an atrophic form, extracellular matrix molecules accumulate in the space outside the RPE, which causes the formation of so-called druses, consisting of fats, vitronectin, amyloid proteins, and inflammatory proteins, accumulated inside the RPE layer (Figure 3). These changes occur in RPE within the maculae region, causing partial cell death, layer disorganization, disruption of RPE functions, and para-inflammatory reaction, which inevitably results in the loss of photoreceptors [26,27]. The wet AMD, also referred to as neovascular (exudative) form, is manifested as the proliferation of a network of blood vessels lining the RPE choroidal membrane in the maculae region. Vessels become dysfunctional and leaky, with fluid and blood accumulating in the maculae region [28] (Figure 3). This causes disjunction of RPE apical processes and photoreceptors, while the connections between them are mandatory for light perception. AMD treatment is diverse, involving neurotrophic factors, growth factors, cell viability factors, and also oxidative stress-preventing factors [2,29,30,31]. With the wet AMD, vascular endothelial growth factor (VEGF) inhibitors and the photodynamic therapy are mainly used [32,33]. Despite efforts aimed at developing adequate therapy, the challenges associated with the treatment of AMD to preserve vision are still substantial. In this regard, the idea of cell replacement with the help of cell sources, including intrinsic ones, to regenerate RPE and photoreceptors becomes highly relevant. The feasibility of such an approach is discussed below.

PVR, often accompanying retinal rupture, is manifested as the withdrawal of RPE cells outside the layer, their epithelial–mesenchymal transition (EMT), transformation, and involvement in the epiretinal membrane (EM) formation [34,35,36]. Proliferative diabetic retinopathy [37] and subretinal fibrosis [38] are also known to be caused by mesenchymal transformation of RPE. In the treatment of this range of disorders, studies of mechanisms and methods for preventing the EMT of RPE cells and the EM formation with their involvement, which prevent normal functional connection between RPE and photoreceptors from being restored, are of particular importance [39].

Retinitis pigmentosa (RP) is characterized as a heterogeneous genetic disorder that leads to progressive devolution of the retina. This congenital disorder has a heterogeneous genetic origin. Approximately 100 genes are known whose mutations may result in RP [40]. It is often accompanied by the loss of peripheral and night vision. This is explained by the initial death of rods, as well as changes in the choroidal network, which, while progressing, leads to the degeneration of cones. Various approaches are currently being developed to slow down the progress of the disorder, including gene therapy, pharmacology, neuroprotection, electrical stimulation, retinal prostheses, and retinal transplantation [41]. Vitamin A preparations and other agents that improve retinal trophy are administered. Since the disorder affects the outer part of the eye—the choroidal coat, RPE, and photoreceptor cells—it would also be relevant to consider the feasibility of replacing damaged cells with genetically healthy ones.

Glaucoma is also one of the major retinal diseases leading to impairment of vision, often irreversible. Glaucoma represents a degenerative optic neuropathy characterized by the progressive degeneration of retinal ganglion cells and the retinal nerve fiber layer. Glaucomatous alterations, often associated with intraocular pressure increase, remain inconspicuous for a long time, but in the final stage, they lead to the death of bodies and axons of neurons, including those in the optic nerve head region [42]. The ocular hypertension and deleterious mechanical forces exerted at the back of the eye, at the level of the optic nerve head/optic disc, are the only modifiable risk factors associated with glaucoma that can be treated. The main approach to treatment is to reduce the intraocular pressure by administering prostaglandins and β-blockers [43,44]. Laser trabeculoplasty and surgery are also used to reduce the rate of disease progression [42]. However, as the death of ganglion cells increases, the necessity arises to replace them with poorly differentiated autogenous RRCSs capable of regenerating the ganglion layer and establishing correct connections with the INL and visual center neurons. 

## 4. Intrinsic Retinal Regeneration Cell Sources and Their Implication for Biomedicine of the Eye

Certain RRCS categories are described in brief in the subsections below. General information is provided in dedicated reviews [3,5,7,8,11,45] and in numerous specialized experimental works. The consideration of each RRCS type is accompanied by an attempt to analyze the prospects for using the accumulated knowledge about RRCSs in biomedicine of the eye.

### 4.1. Retinal Ciliary Zone Cells

RRCSs of this category are located on the extreme periphery of the retina, in the so-called corner of the eye (Figure 1). In vertebrates, the ciliary zone cells exhibit pronounced, to varying degrees, features of stemness/early progenitors. There are differences in the size of this cell population, decreasing in the series from fish to mammals. In fish and amphibians, persistent neurogenesis occurs in the poorly differentiated ciliary marginal zone (CMZ) that provides recruitment of cells to the growing retina [46,47]. The cells of this zone are also involved in the regeneration of the retina in case of damage [46,48,49,50]. 

CMZ cells of lower vertebrates have been studied as regards their internal molecular genetics and epigenetic profiles, and also functions of the signaling microenvironment. Expression of transcription factors (TFs) responsible for neurogenesis and regulatory signaling pathways that activate and/or block neurogenesis in the CMZ has also been investigated [3,51,52,53]. Eye field TFs in development belong to those whose expression is characteristic of CNS cells and causes manifestation of their neurogenic potential [54,55]. Their spectrum is encoded by members of the homeobox gene family: *Pax6* and *Rx*, *sine oculis*, *Six3*, and *Lhx*. Their list includes the genes *Chx10* (*vsx2* in fish), *Crx*, and *Pitx*, being also active at early stages of eye development [56,57].

Basic information about the CMZ has been collected through studies on the retina of fish [51,58,59] and amphibians [46,60] in development and regeneration. In these animals, the CMZ is composed of peripherally located stem cells and more centrally localized progenitors. The former are capable of continuous independent divisions; the latter divide only to a limited, regulated number [51,61,62,63]. The *Xenopus laevis* model has shown that the expression of genes encoding eye field TFs is characteristic of the most peripheral part of the CMZ. In *X. laevis,* Rx^+^ cells of the CMZ prior to metamorphosis are involved in retinal regeneration after partial incision; the *Rx* knockdown could impair regeneration by leading to a drastic reduction in proliferation [64]. The expression of *Rax* downstream genes was detected in the CMZ of *X. laevis* mature retina, which indicates the progenitor properties of these cells [65]. A study of gene expression in CMZ cells of mature caudate amphibians has also shown the progenitor characteristics of cells of this zone, which is involved, along with RPE, in epimorphic NR regeneration in these animals [49]. Expression of TFs Pax6, Otx2, Six3, and also Prox1 and Pitx2 was recorded for this zone in newts *Pl. waltl* [66,67,68]. In order to increase the neurogenic potential of the CMZ, attempts have been made to enhance the cell proliferation and differentiation in this zone using fish and amphibian models. This can be exemplified by the ability of CMZ cells to maintain a high level of proliferative activity, as has been reported for *dmbx1a* germline zebrafish mutants [69]. It has also been shown that the activation of the IGF receptor gene (*Igf1r*) in medaka causes a decrease in the cell cycle length and an increase in the production of differentiated neurons [70].

In reptiles, birds, and mammals, this retinal region is reduced, however, with few cells still retaining signs of a low level of differentiation and proliferation ability [50,71,72,73]. The CMZ in chicks before hatching is similar to that of adult fish and amphibians, but its contribution to retinal growth is small [74,75]. After hatching, the CMZ in birds does not disappear completely, and the neurogenesis remains extremely limited [76,77,78,79]. An attempt has been made to obtain neurons in vitro using cells from the ciliary margin of the chicken retina, whose neurogenic potential in vivo is very low [80]. Cultivation of cells isolated from the ciliary margin caused them to form so-called neurospheres. The neurosphere cells exhibited apicobasal polarity and acquired positional values along the radial and tangential axes close to those observed during NR development in vivo. When the neurospheres were placed in differentiation-promoting media, the cells acquired the phenotype of ganglion cells, forming long processes. In the latter, the key regulators of axon growth including, in particular, Eph/ephrin were identified [80]. The study showed for the first time the potential for reproduction of autologous, differentiated ganglion cells from the ciliary margin of the chicken retina in vitro. Though carried out on an animal model, the study provides a basis for attempts to reproduce human NR ganglion cells in vitro for transplantation purposes.

In mammals, the CMZ represents a very small number of cells localized in the area of transition from the retina to the ciliary body (CB) at the extreme periphery of the NR. In humans, it is a non-laminated part of the NR, also referred to as *ora serrata* [81]. Normally, these cells do not show proliferative activity and, as was previously suggested, mammals should lack any analogue of CMZ. However, a study of NR ciliary margin cells in mouse embryos revealed a limited neurogenesis with ganglion cell formation [82]. It is also known that cells of the marginal region of mammalian and human NR in early development express such TFs as Pax6, Chx10, Lhx2, Otx1, Prox1, Pitx1/2, retinol dehydrogenase Rdh10, and other markers, thus indicating a low level of differentiation of these cells [83,84,85]. In prenatal mice, cells of the NR marginal zone are positive towards BMP4, the Cyclin D2 proliferation factor, and also the TFs Msx1 and Zic1/2 [82,85,86]. There is evidence that in one of the adult mouse lines, cells of the NR marginal region retain the neurogenic potential and express *Atoh7*, a marker gene of neurogenesis [72]. The expression of *Atoh7* is assumed to be an autonomous response of the NR ciliary margin cells to apoptosis of ganglion cells. With the targeted elimination of part of the ganglion cells, these cells implement the neurogenesis potential, which results in the recruitment to the ganglion cell population [72]. There is also evidence of the proliferative activity of cells of the NR ciliary epithelium in rats with experimentally modeled RP [87]. Authors of this study explored the proliferation potential of ciliary marginal cells of Royal College of Surgeons (RCS) rats, an animal model for RP. A few Chx-10 and BrdU labeled cells were found, and their number significantly increased on day 30 and 60 of RCS postnatal development. 

These results suggest that the cells in the proliferating marginal region of the mammalian retina have the potential to regenerate the retina following its degeneration caused by RP [87].

Spontaneous migration of NR *ora serrata* cells, isolated in human explants and cultured in vitro, has recently been detected [88]. Co-expression of proliferation markers and Müller glial cell markers has also been observed in migrated cells. Proliferation of the NR ciliary epithelium cells with the subsequent acquisition of neuronal/terminal differentiation by them has been described for cases of retinal detachment and PVR in adult humans [89]. These observations suggested that the epidermal growth factor receptor (EGFR)-positive ciliary epithelium cells could form “neurosphere-like” structures and their differentiation was directed towards the neural and photoreceptor (but not glial) lineage formation. Therefore, in the adult human eye, the ciliary epithelium in a pathological retinal environment such as retinal detachment and PVR may provide a spontaneous source of donor cells for retinal transplantation [89].

These data indicate that the ciliary margin cells of the mammalian and human NR in embryonic development retain some features of progenitor cells. Under targeted experimental conditions or NR pathology in vivo, cells of the ciliary region are capable of proliferation and production of retinal cells. Understanding the molecular basis and searching for such conditions are among the promising objectives of regenerative biomedicine of the eye, aimed at reproducing ganglion, glial or other phenotypes of retinal cells to replace dysfunctional or lost ones.

In this regard, information about the external regulators of RRCS behavior in the CMZ is of particular importance. The major signaling regulatory molecules in the microenvironment of CMZ cells are IGF, Shh, Wnt, Notch, and glucagon [69]. In the NR development, CMZ cells, against the background of Notch activation and in its interaction with the TF Atoh7, are responsible for the production of bipolar, amacrine cells, and MGCs. A certain balance between Notch and Atoh7 is considered as a basis during maturation of the range of neurons produced by the CMZ in the postembryonic period [90]. In *X. laevis*, the Shh ligand, produced by ganglion cells, accelerates the cell cycle exit by suppressing cell proliferation in the CMZ. The effect of Shh signaling is exerted through the activation of cyclins, with the reduction in the length of the G_1_ and G_2_ phases of the cell cycle [91]. In chicks, overexpression of Shh in Pax6-positive CMZ cells induces expression of *Gli3/Gli1*, mediators of Shh signaling, which, in turn, enhances proliferation in the CMZ [92]. The factors activating proliferation in the CMZ are the Wnt ligand or expression of its effector genes found in mouse, chicken, frog, and fish [93,94,95,96]. In mouse, the activity of Wnt signaling is suppressed by the TFs Six3/Six6. In the ciliary marginal region of the retina, the *Six3/Six6* expression is reduced, which leads to up-regulation of the Wnt signaling pathway that supports the progenitor characteristics of cells [97]. Glucagon is also a factor regulating the proliferation of CMZ cells. In the chick NR, glucagon is normally produced by one of the populations of amacrine cells (bullwhip amacrines), extending their long processes to the CMZ and suppressing the proliferative activity with glucagon there [98]. 

There is information about the epigenome of CMZ cells obtained using animal models. Angileri and Gross [99] studied DNA methylation in CMZ cells by investigating the work of the Dnmt1 methyltransferase catalyst. The role of expression of this gene in the maintenance of progenitor properties by CMZ cells was determined using zebrafish mutants with knocked out *dnmt1*. The *dnmt1*-deficient fish (*dnmt1^−/−^)* showed a decrease not only in the number of cells in the CMZ, but also in their proliferative activity. The authors suggest that the *dnmt1*-related epigenetic modification in lower vertebrates is responsible for maintaining the poorly differentiated state of CMZ cells [99]. Histone acetylation plays a certain role in neuroprotection in the CMZ during neurogenesis. Histone deacetylase inhibitors (HDACi) can protect CMZ cells from death in mutant (*dying on edge*, *dye*) zebrafish that have mass cell death in this zone [100]. To date, all of the hypothesized variety of epigenetic mechanisms responsible for and regulating the progenitor status of cells and neurogenesis in the CMZ during the NR development and regeneration requires in-depth study.

The obtained data on CMZ cells were considered by Miles and Tropepe [69] in association with microphthalmia. This genetically inherited disorder is related to a deficiency in progenitor cell proliferation in this region of the eye. Myopia, a well-known vision condition, is also associated with the growth and shape of the eye that, in turn, are related with the cell behavior in the retinal growth region [101].

Taking into account the collected information about the extremely limited potentials of the mammalian NR ciliary zone, any natural formation of new neurons or RPE cells in this zone in humans is improbable. Nevertheless, the extensive information about the genetic properties of ciliary region cells and the regulatory molecules of their progenitor state, proliferation, and neurogenesis can potentially be applied in biomedicine of the eye in the future. Furthermore, this knowledge contributes to the development of therapies for not only retinal diseases directly associated with developmental CMZ disorders, but also diseases requiring cell replacement. Reproduction of ganglion cells in vitro, as shown in the above-cited studies by Fiore et al. [80] on a bird model, can be developed using material from embryonic and adult rodents. Cells with progenitor properties and Müller glial cells can be produced by cultivating fragments of tissue from the same area of the retina that are obtained during eye surgery in humans and usually discarded [88].

### 4.2. Ciliary Body Cells

As mentioned above, in adult mammals, the region similar to the CMZ is extremely reduced by the number of cells and does not normally exhibit any regenerative abilities [102]. A region, close in localization but not analogous to the CMZ of lower vertebrates, in adult mammals and humans is represented by the ciliary body (CB) (Figure 1). The CB in the mammalian eye has two cell layers and muscles. The outer pigmented layer is a continuation of the RPE; the inner, non-pigmented layer, a continuation of the NR. The cells constituting the CB have a specialization different from that of RPE and NR neurons. They produce components of vitreous fluid and are involved in visual accommodation [103,104]. However, in case of damage to the NR leading to the loss of ganglion and amacrine cells, cell proliferation is known to be activated in the non-pigmented CB layer [6,102,105,106,107]. In adult mice, some CB cells re-enter the cell cycle and change their phenotype, expressing TF Chx10 and also marker proteins of bipolar and photoreceptors as a response to the optic nerve transection causing the death of ganglion cells [105]. Expression of retinal progenitor genes was observed in the adult mouse CB after intraocular injections of regulators of Rho GTPase activity [108]. The injections enhanced the co-expression of TFs Pax6 and Chx10, but showed no effect on proliferation in the CB. The inactivation of Rho GTPases conversely increased the proliferation of CB cells, including those exposed to growth factors. The authors suggest that the approach to and understanding of the ways of regulation of CB cell proliferation and differentiation can be used to replace dead ganglion and photoreceptor cells with CB cells [108].

Numerous in vitro studies have shown that human, primate, pig, rodent, and chicken CB cells can express Nes, Mitf, Pax6, Six3, Rx, Chx10, and FGF2, which are markers of stem cells and retinal progenitors [109,110,111,112,113]. In a number of studies [109,114,115], cell aggregates (neurospheres) were obtained from dissociated mouse and human CB cells. As the authors assumed, these aggregates were formed by multipotent cells. The cells within the neurospheres proliferated, were nestin (Nes) and Pax6 positive, and simultaneously expressed Claudin-1, a marker protein of epithelial cells. Cells that expressed protein markers of retinal neurons, in particular, photoreceptors, were found in rare cases. After further investigating the issue, Cicero et al. [116] concluded that CB cells are not truly stem cells, as they retain certain features of pigmented epithelia. Such results and the low production of retinal precursors by descendants of CB cells in neurospheres in vitro have reduced interest in CB as a potential cell source to replace dying NR neurons [117]. However, attempts are still being made to increase the production of neurons from CB in vitro by manipulating cultivation conditions [118,119,120]. In a study on cultures of ciliary/CB epithelium cells of postnatal pigs [121], expression of pluripotency marker genes *Klf4*, *Sox2*, and *cMyc* was detected in a suspension of cells derived from neurospheres. After the latter were dissociated and labeled with CM-DiI vital dye, the cells were injected subretinally into the eyes of adult mice. After transplantation, the cells acquired a mixed phenotype, showing features of retinal neurons and RPE cells. Some of them were incorporated in the RPE, where multilayered RPE65^+^ cell loci were observed. Another portion of the cells (5–10%) expressed marker proteins of retinal neurons: recoverin, protein kinase C, and calbindin [121]. These results directly indicate the expediency of continuing the studies on CB cell potencies to be used for cell replacement in the treatment of retinal degeneration.

Little is known about the external regulators of CB cell reprogramming in vitro. As early studies [122,123] showed, when Notch signaling is disrupted, the formation of neurospheres from CB cells is blocked. Notch has recently been discovered to play a role in the development of CB, where combinations of Notch family proteins are necessary for morphogenesis, differentiation, and the establishment of CB functions [124]. Thus, Notch 3 in cooperation with Notch 2, regulating the adhesion factor Nectin1, is responsible for the CB morphogenesis. Pang et al. [124] suggest that Notch regulation may underlie the progression of glaucoma. Signaling including the tyrosine kinase receptor, c-Kit transmembrane protein and its ligand, stem cell factor, are also assumed to be involved in the regulation of reprogramming of CB cells forming neurospheres in vitro. These components are capable of supporting not only the proliferative activity of cells in CB-derived spheres, but also their differentiation in the retinal direction [122,123]. 

In a study of CB-derived neurosphere cells in vitro, some epigenetic regulators of retinal cell differentiation were identified. For this, Jasty and Krishnakumar [125] analyzed DNA methylation and histone methylation-H3K4me3 and H3K27me3 in a CB-derived lineage committed progenitor to terminally differentiated cells isolated from the CB of human cadaveric eyes. The authors detected bivalent modifications involved in the process of differentiation of stem/progenitor cells into neural and glial cells [125].

Thus, the mammalian, including human, CB contains cells capable of acquiring retinal differentiation. These are few in number and their initial differentiation in vivo is stabilized, which is consistent with the functional destination in the CB. However, studies of molecular genetics and epigenetic profiles, as well as the potential of reactivation to proliferation and genesis of cells with signs of retinal differentiation revealed under experimental conditions, still indicate CB cells as potential candidates for application in cell technologies. In addition to the accumulated knowledge that can be used to obtain an in vitro cell resource for replacing RPE and dead NR neurons, the data on the biology of CB cells and its regulation can contribute to the development of glaucoma therapy [42].

### 4.3. Retinal Pigment Epithelium Cells

In adult vertebrates and humans, RPE is a monolayer of pigmented, epithelial, and specialized cells. The RPE is oriented towards the NR with its apical side; on the basal side, it is limited by the Bruch’s membrane and the vascular membrane referred to as choroid (Figure 1 and Figure 2). The RPE is multifunctional: apart from transferring substances from the choroid to the NR, it protects against oxidative stress, produces growth factors, and metabolizes vitamin A derivatives. A major RPE function is phagocytosis of the outer segments of photoreceptors, their digestion by lysosomes, and retinoid metabolism, i.e., providing the processes required for light perception [126,127,128,129].

Among the well-known RRCSs, RPE has been studied to a greater extent, including the possible implications for practical use. Damage to the RPE layer and its cells, and also disturbance of their relationship with photoreceptors, are the causes of most degenerative diseases of the retina. In this regard, and taking into account the common origin of RPE and NR in the development of the eye and the possibility of their mutual conversion [39], RPE has been extensively studied using animal models and in humans. Studies have been conducted in various directions: cell functions, differentiation of RPE and its changes during regeneration, and congenital and acquired eye pathologies; RPE behavior in vitro and under transplantation conditions has also been investigated [127,130,131,132,133,134,135,136].

A study on a model of zebrafish larvae and adults has shown that RPE regeneration is possible even in the case of significant damage to the layer. To develop an RPE injury model, a transgenic line (*rpe65a*:nfsB-eGFP) of fish was used [137]. The regeneration of the RPE layer occurred through proliferation of cells adjacent to the area of damage, compensating for the loss of the layer. Based on transgenic and mutant zebrafish lines, pharmacological manipulations, and transcriptomic and morphological analyses, the authors investigated the immune response mediating the regeneration of the RPE layer. They found that RPE cells express the immune response genes, in particular, interleukin 34, and that macro- and microglial cells are necessary to maintain this immune response [138]. 

RPE of caudate amphibians (Urodela) is a classic example showing the potential for natural regeneration of the RPE layer along with its reprogramming into NR cells in mature animals in vivo. Even in the case where the original NR is removed in newts, the RPE becomes a source of a new, functioning NR and, simultaneously, restores its own layer [133,139,140,141,142,143]. During NR regeneration, RPE cells leave the layer, lose the initial features of specialization, proliferate, and form an intermediate population of cells with neuroblast properties. After six or seven cycles of cell divisions increasing the population of the NR regenerate, neuroblasts acquire phenotypes of retinal neurons and glial cells and begin to function. The initial number of cells in the RPE layer is restored through the proliferation of cells retained in it. The process of newt RPE reprogramming, the NR anlage formation, and its differentiation are regulated by TFs, signaling molecules, and epigenetic controllers. After the disconnection of RPE and NR, the genes of the immune response and proto-oncogenes (*c-fos*, *c-myc*, and *c-jun*) are activated first [144]. It was also found that the daughter cells of RPE at the beginning of retinectomy-induced proliferation express the pluripotency genes *c-myc*, *Klf4*, and *Sox2* and, along with them, developmental *Mitf* and *Pax6* [145] and the neural stem cell marker Musashi-1 (Msi-1) [146]. Expression of genes characteristic of eye development (*Pax6, Prox1, Six3, Pitx1,* and *Pitx2*) along with tissue-specific *RPE65* and *Otx2* has also been revealed [66,68,142,147,148]. The patterns of expression of both marker and regulatory molecules in NR development and regeneration are largely similar. In an experiment by Casco-Robles [149], RPE cells acquired mesenchymal-like phenotype, and NR regeneration was blocked. This was caused by the knockout of the *Pax6* gene in larval newt *Cynops pyrrhogaster.* Leaving the RPE layer, these cells formed aggregates showing the expression of myofibroblast proteins (a-SMA, Vim, and N-Cad). These data are important for understanding the role of the Pax6 gene in the mesenchymal transformation of RPE cells that occurs in cases of PVR and retinal fibrosis in humans [34,35,36]. 

Studies of regulation of RPE cell differentiation by the microenvironment also provide clues to the directed reprogramming of these cells. Among the signaling cascades controlling NR regeneration in Urodela and zebrafish, special attention is paid to the Fgf, Bmp, Wnt, Shh, and Notch signaling pathways [150,151,152,153]. The key role of FGF2 in NR regeneration in newts (as in other animals) has been identified [154,155,156]. It has been found that FGF2 is not the primary trigger of RPE reprogramming, but is required for regulating *pax6* gene expression at the onset of the process and then for increasing the proliferation and entering the differentiation of RPE-derived cells [152,157,158]. As for other RRCSs of the eye, the neural conversion of RPE in salamanders depends on Notch signaling. According to [150] and [151], the introduction of the DAPT blocker induces premature maturation of neurons in the NR regenerate. The mechanism of epigenetic regulation of the RPE reprogramming into NR cells in salamanders has not been studied, but the permissive factors of the epigenome are assumed to be the expression of pioneer TFs (opening repressed chromatin domains) and demethylation of regulatory elements of photoreceptor genes [159].

As in newts, RPE potencies to regenerate NR through reprogramming have been found in the frog *X. laevis* [160,161]. The up-regulation of the “developmental” *pax6* and *rax* has been shown for the reprogramming of cells derived from RPE of larval *X. laevis*. Knockdown of the *rax* gene inhibits the formation of specific cell types in the NR regenerate [162,163]. The expression of the *rx* gene is necessary for the formation of NR regenerate in pre-metamorphic *X. laevis* [64,164]. The key external regulator of RPE conversion in *X. laevis* is FGF2 promoting the proliferation [161,165]. Inhibition of the FGF2-dependent MAPK signaling pathway reduces the size of the NR cell population forming the regenerate [161]. Matrix metalloproteases show positive regulation of the NR regeneration from RPE cells and are, in turn, regulated by the inflammatory cytokines IL-1β and TNF-α [166]. The model of NR regeneration from RPE in *X. laevis* is reported to allow a wide range of applications of molecular genetics methods to identify the mechanisms triggering the activity of regeneration-associated genes [167]. 

In chicks, NR regeneration from RPE cells occurs at early ontogeny stages (up to E4–E4.5) [168,169]. A fragment of C3a complement is considered to be an inducer of the NR regeneration process in chicks. This short-lived polypeptide activates STAT3, enhancing the action of IL-6, IL-8, and TNF factors, which leads to up-regulation of the Wnt2b signaling genes and expression of the *Six3* and *Sox2* genes characteristic of retinal progenitor cells [170]. A study on an embryonic chick model (stage E4) [171] has revealed nuclear β-catenin as an obstacle to the entry of RPE cells into the cell cycle. In the presence of FGF2, its expression was lost, thus, indicating the inactivation of nuclear β-catenin as one of the conditions for NR regeneration from RPE cells in chicks [171]. The involvement of Shh [172], BMP, and Wnt signaling cascades [173] in NR regeneration has been reported for birds as well. It is also known that during the formation of the RPE and NR domains in the optic cup of the chick embryo, a low concentration of BMP leads to the RPE to NR conversion, whereas a high one conversely leads to the NR to RPE changeover [173]. Along with BMP, Wnt signaling plays an important role in such transformations. The authors suggest that the control of these two key signaling pathways can provide the basis of a protocol for the production of RPE and NR cells for cell replacement. In the study by Tangeman et al. [174], laser capture microdissection was used to isolate RNA from intact RPE, transiently reprogrammed RPE (t-rRPE) 6 h post-retinectomy, and reprogrammed RPE (rRPE) 6 h post-retinectomy with FGF2 treatment. RNA sequencing of individual cells (scRNA-seq) and a comparative analysis of transcriptomes were carried out. At an early stage of RPE conversion, up-regulation of damage-associated genes and repression of genes responsible for the entry and progression of the cell cycle were detected. In the presence of FGF2, on the contrary, the level of expression of MAPK activation genes was high, which confirms the assumption that the FGF2/MAPK signaling cascade is the main driver of RPE conversion in embryonic chickens [174]. An attempt was made to understand the mechanisms of the EMT strategy of mammalian RPE cells in PVR or fibrosis using the same animal model. It turned out that the expression of EMT-associated genes was suppressed at the onset of the RPE conversion, but later activated. The expression of EMT-associated genes (*SNAI1*, *TGFBR2*, *ELK3*, *SMAD3*, and *TGFB3*), presumably dependent on extracellular matrix, is modulated during these events. Thus, parallel studies of the expression of genes responsible for the neuronal conversion of RPE cells and the genes responsible for EMT can answer the question as to why RPE of avian embryos avoids mesenchymal transformation [174].

Variations in the phenotype and proliferative activity of RRCSs occur with the involvement of the changing epigenetic landscape that modifies the transcription program. Using an embryonic chick model, Luz-Madrigal et al. [175] carried out TAB-seq and ChIP-seq to study the process of NR regeneration from RPE. RPE cells were studied prior to retinectomy and at various RPE reprogramming stages. The data indicated a significant rearrangement of the DNA methylation pattern. Regions of differential methylation of gene promoter sites, associated with chromatin organization and FGF2 production, were identified. The study highlights the implication of DNA demethylation in RPE cells for overcoming epigenetic barriers to NR regeneration from RPE in mammals and humans [175]. Thus, the competence of RPE cells in amphibians and birds to be reprogrammed into retinal cells and the mechanisms of their implementation are based, like in other animals, on the extracellular and intracellular features of molecular regulation of cell behavior. Understanding of these facts, taking into account the conservatism of the main tools to regulate the differentiation and proliferation of RPE cells in animals and humans, is of fundamental importance for the development of biotechnological approaches aimed at regeneration of damaged human retina.

In mammals and humans, RPE layer regeneration is extremely limited: only very small damages can be repaired through expansion of neighboring cells. An exception to this rule is mutant MRL/MPJ mice, in which repair of minor damage to RPE is possible on the periphery of the layer [176]. A study by Kampik et al. [177], carried out also on genetically modified mice, showed the probability of partial RPE regeneration with the lack of cell transformation into neural or other phenotypes. According to our observations during experiments with organotypic 3D cultivation of the posterior sector of the rat eye, the RPE layer under conditions of partial death of its cells still continued to maintain the epithelial structure for a long time. This occurs through the increase in the size of cells and their stretching over the layer; however, such RPE cells cannot establish a normal functional connection with photoreceptors, simultaneously undergoing death [178]. The models of in vivo RPE damage in mice showed an increase in the size of RPE cells and also in the proportion of multinucleated cells [179,180]. When damaged or disconnected from the NR, the RPE is known to produce a wide range of factors, including PEDF, BDNF, and CNTF [181,182]. These RPE responses are of reparative nature and aimed at maintaining the viability of cells and their integration with the NR. The RPE in mammals, including humans, is not capable of regeneration, despite the fact it has rare cell divisions detected on the periphery of the layer [134,183,184]. 

In vivo proliferative activity and depigmentation of RPE cells are known for a number of pathological conditions of the retina. Proliferation of RPE cells accompanies the above-mentioned vitreoretinopathy (PVR). In the first stage of the disease, often associated with NR detachment, a portion of RPE cells lose their pigment and epithelial properties, leave the layer, migrate beyond the NR, and proliferate (Figure 4). These processes are accompanied by the death of RPE cells or their conversion into myofibroblasts. During migration, while exposed to the surrounding conditions, RPE cells proliferate, initiate the synthesis of extracellular matrix components, and are involved in epiretinal membrane (EM) formation [185,186,187,188,189]. The formation of EM and its contraction along with NR are responsible for the clinical manifestation of PVR. The key event of PVR is the epithelial–mesenchymal transition (EMT) of RPE cells [35,36,190,191]. Note that the EMT occurs not only in PVR, but also in the case of the RPE attempting to recover and restore the layer after laser-induced damage [192], subretinal fibrosis [193,194], or proliferative diabetic retinopathy [37]. The EMT is manifested as the loss of the polarity of RPE cells, destruction of their contacts, and detachment from the Bruch’s membrane. These events are accompanied by the cytoskeleton reorganization and acquisition of a mesenchymal phenotype [195,196]. A reorganization of cell–cell contacts occurs when the expression patterns of cadherins and associated catenins change [197,198]. Smooth muscle alpha actin (α-SMA) is an intracellular protein, a marker of transitory epithelial–mesenchymal differentiation that provides mobility of RPE cells. Vimentin is responsible for stabilizing the structure of RPE-derived cells during migration [199]. As EMT progresses, the pattern of cytokeratin expression also is altered, and extracellular matrix proteins, including collagen and fibronectin, are deposited [198]. 

Variations in the expression of functionally and structurally significant genes under the control of TFs, epigenetic factors, and external regulatory signaling systems constitute the molecular genetic basis for the EMT of RPE cells [200,201]. The role of the FOXM1 proto-oncogene as a regulator of EMT and RPE cell proliferation has been identified [202,203]. The expression pattern of TFs belonging to the Snail and Slug, ZEB1/2, TWIST, GSC superfamilies [204,205] and other TFs usually accompanying EMT in fibrous or onco-transformation is known to undergo alteration with the EMT [206,207]. The TFs that control the key EMT mechanism—the initiation of expression of E-cadherins—play a major regulatory role. This range of TFs is represented by Oct-1, hepatocyte nuclear factor 1 (HNF-1), such TFs as GATA-1, SMAD3, and TFE, interferon regulatory factor-1 (IRF), etc. [205,208].

Micro RNAs (miRNAs) are also involved in the regulation of gene expression in the PVR process [209,210]. Maps of the associated epigenetic and transcriptional changes in human RPE with simulated EMT, compared to the normal ones, have been reconstructed. Active enhancers and TFs associated with actively transcribed genes have been identified in studies of the RPE epigenome and transcriptome in normal conditions and after treatment with key positive EMT regulators, TGF-β1 and TNF-α. In parallel, nicotinamide (NAM) has been found to be able to suppress EMT-associated key transcription events in human RPE cells [211]. The external mediators of the EMT process in PVR are growth factors and inflammatory factors. The process is initiated by TGF-β [212] and develops with the involvement of TNF-α, PDGF, EGF, FGF, VEGF, CTGF, IGF2, IL-1a,β, IL-2,3,6,8, adhesion factor ICAM-1, and other signaling [205,213,214,215,216]. An increase in the TGF-β level was observed in vitreous bodies of patients with PVR, which correlated with the severity of the disease [217]. TNF-α enhances the expression of genes associated with apoptosis and cell motility in RPE [218]. Both factors synergistically activate the EMT program in adult RPE cells. Modulation of RhoA/Rho-kinase, Smad, or MAPK signaling is considered among the strategies for suppressing signaling pathways involved in the RPE pathology [212,219]. Exposure to the above-mentioned NAM can enhance the epithelial phenotype of RPE and prevent the EMT. For this reason, NAM is considered an agent for preventive treatment of RPE/retinal pathologies [211,220]. Other approaches involve blocking the expression of one of the TGF-β receptors such as, in particular, activin receptor-like kinase 5 (ALK5), which facilitates suppression of the EM formation [221,222]. The causes and progression of PVR, resulting from genetic aberrations that alter the behavior of RPE cells, are the subject of further molecular research to provide sufficient data for biomedicine of the eye. 

It seems that the impossibility of restoring the RPE layer and regenerating NR from it should discourage researchers from considering RPE as a potential RRCS in mammals including humans. However, data on RPE cells’ behavior in vitro not only revive these hopes, but also contribute to understanding of the RPE-associated retinal diseases. The data from in vitro experiments clearly indicate the plasticity of the mammalian RPE phenotype and the potential of de- and re-differentiation, and also of reprogramming into other retinal cell phenotypes. Rodent and human RPE cells exposed to morphogens and growth factors in the culture medium lose their original properties and proliferate [223,224,225,226]. Adult rat RPE cells in vitro express marker proteins of neural progenitors: Nes, Musashi1, doublecortin, and β-III tubulin [224]. As was shown on mouse embryos, the neural conversion of RPE is blocked by activin, one of the TGFβ superfamily proteins [227]. A series of studies of isolated human RPE cells and cell lineages in vitro revealed a decrease in the level of specific differentiation and expression of RPE65 along with the activation of the progenitor cell marker genes *OCT4*, *NANOG*, *KLF4*, *OTX2*, *PAX6*, and *NES* [225,228,229]. In some cases, the expression of pan-neural markers, tyrosine hydroxylase and neurofilament proteins was recorded [225]. The introduction of FGF2 into the medium caused RPE cells to exhibit proneural properties [230]. Analyzing the behavior of RPE cells depending on their cultivation conditions, Burke [231] found a relationship of the phenotype modulations of RPE cells with the Wnt/β-catenin signaling. There is evidence of the role of the highly conserved Hippo signaling in this process [232]. It has also been reported that mouse RPE cells forming neurospheres in vitro represent a population whose cells are capable of returning to the original phenotype or acquiring photoreceptor differentiation. When subretinally transplanted into mice with retinal degeneration, RPE-derived neurosphere cells could integrate into the RPE and NR, thereby delaying retinal degeneration. This finding clearly shows the perspectives for the use of mammalian RPE cells to replace dying cells in NR diseases [232]. 

Study of the mesenchymal conversion of human RPE cells and its regulation in vitro is of certain interest for biomedicine in the context of PVR therapy. A study by Salero et al. [233] showed that human RPE cells in vitro can reproduce cells expressing markers of mesenchyme-derived cells: muscle, adipo-, osteo- and chondrogenic cells. This occurs where the components stimulating the manifestation of these differentiations are added to the medium. The Wnt signaling pathway plays a significant role in the regulation of mesenchymal differentiation in the ARPE-19 cell line in vitro [234]. Based on the behavior of mammalian and human RPE cells during in vitro cultivation, some authors [232,233,235,236] consider them (or a small number of them in the RPE layer) stem cells. A recent study by Pandey et al. [237], based on the results of scRNA sequencing, revealed the molecular features of heterogeneity in the adult murine RPE. Approximately 1–2% of the total number of cells exhibited the hallmarks of stem and/or progenitor cells. The authors suggest that RPE stem cells may also exist in the human tissue. 

Compared to the data on the potential of the neural and mesenchymal conversion of the human RPE, the information on ways to keep RPE cells from reprogramming and stabilize their initial phenotype through signaling molecules in vitro is no less important. Studies of transcriptomes of such RPE cells maintaining initial differentiation in vitro indicate that they retain properties close to native ones [238,239,240,241].

Thus, the results of the studies on RPE as a source of retinal regeneration/degeneration in animals and humans in vivo and the behavior of mammalian RPE cells in vitro make an undoubtedly substantial contribution to biomedicine of the eye. This contribution is determined by the observed ability of RPE to produce retinal neurons and by the identification of external and internal mechanisms of RPE cell phenotype regulation. The former allows obtaining autologous cells in vitro to replace dying cells in the NR. The latter makes it possible to control the behavior of RPE cells to prevent their death and undesirable phenotypic transformations, design methods for the treatment of retinal diseases, and also study the molecular genetic basis of such diseases.

### 4.4. Müller Glial Cells

Müller glial cells (MGCs) are well-known and widely studied as latent RRCSs (see reviews [242,243,244,245,246]). According to data obtained recently, MGCs that have undergone age-related changes can still be stimulated to regenerate cells lost after acute NR damage in aged zebrafish [247]. MGCs are very promising for regenerating cell losses in the NR [8,245,248,249]. 

MGC bodies are located radially in the INL, extending long processes to the outer and inner limiting membranes of the NR (Figure 2). The Müller glia is a cell population specialized in performing a wide range of functions, including neurotrophic and structural ones, and also maintaining synaptic connections with NR neurons. It is involved in both NR cleaning and light perception [250]. Furthermore, extensive evidence indicates that MGCs are a population that can exhibit the properties of neural progenitor cells. If NR is damaged, they re-express TFs (six3, pax6, rx1, olig2, and vsx2) characteristic of NR progenitors and immature macroglial cells [251]. In the case of surgical excision of zebrafish NR, MGCs proliferate and produce retinal progenitors capable of differentiating into photoreceptors (cones) and interneurons [252]. After thermal or light-induced damage to the fish NR causing loss of photoreceptors, MGCs re-enter the cell cycle and subsequently up-regulate the expression of stem and progenitor cell-specific proteins [51,253,254]. During the life-long growth of the eye in fish, MGCs maintain the rod photoreceptor lineage, and in case of regeneration, they are able to produce precursors for photoreceptors and ganglion cells [242]. Emerging evidence shows that inflammation plays an essential role in the multi-step process of retinal regeneration [255]. The zebrafish model, with its extensive experimental manipulation capabilities, has been accepted for the study of MGC reprogramming in order to stimulate MGC conversion in mammals [256] (Figure 5).

In birds after hatching, MGCs exposed to the growth factors IGF and FGF2 also demonstrate regenerative responses: they proliferate and express “developmental” TFs [77,257,258]. The transcriptome of MGCs in postnatal mammals shows homology with that of dividing NR progenitors during ontogeny. After the first postnatal week in mice, 68% of the genes specifically expressed in MGCs match those of proliferating progenitors, and only 14% are associated with photoreceptor differentiation [259]. Differentiated MGCs of the mouse retina are characterized by the expression of genes necessary for proliferation [260]. In the normal adult NR in vivo, MGCs express CD117 and CD44, which are marker proteins associated with stem cells, and, simultaneously, vimentin, a marker of glial differentiation [261]. It is found also that adult human MGCs express markers of neural progenitors including SOX2, PAX6, CHX10 and NOTCH, and become spontaneously immortalized in vitro [262]. Authors suggest that MGCs constitute a potential resource of retinal neurons for transplantation studies. 

Simultaneously, mammalian retinal MGCs respond to damage by reactive gliosis, manifested as up-regulation of stress proteins, proliferation, hypertrophy of MGCs, and strengthening of the radial glia differentiation traits [263] (Figure 5). Such a behavior of MGCs is characteristic of many pathological conditions of the NR accompanied by the death of neurons. Reactivation of MGCs is aimed at protecting the NR tissue from destruction and supporting its functions through the release of antioxidants and neurotrophic factors [264]. The extent of neuronal death determines the proliferative activity of MGCs in mice, while the factors secreted by neurons provoke MGCs to activate the EGFR-ERK pathway necessary for proliferation. This, in turn, suggests that cell death-associated signaling pathways may be considered a therapeutic target to prevent proliferative gliosis in NR degeneration [265].

MGCs of the mature NR proliferate, not only in conditions of NR pathology in vivo, but also when cultivated in vitro [266,267]. Components of the EGF, FGF, Notch, Wnt, and Shh signaling pathways, when exogenously added to the culture medium, can provoke mammalian MGCs to exhibit the properties of retinal progenitors [257,268,269,270,271,272]. Data obtained on human MGCs also indicate the genetic traits characteristic of RRCSs, i.e., a combination of genes associated with both NR progenitors and functional specialization genes [45]. MGCs obtained from the NR of cadaveric human eye samples and cultured in vitro under exposure to growth factors showed a decrease in the level of differentiation, acquiring the phenotype of proneural precursors [273,274]. Taking these features into consideration, it seems relevant to search for regulators that allow or block gliosis and reprogramming of mammalian, including human, MGCs.

MGCs are also related to the diabetic retinopathy, in which they dysregulate the neuronal function and produce proangiogenic and pro-inflammatory factors, thus, creating an environment facilitating neuronal dysfunction [275]. In this regard, restoration and maintenance of normal MGC functions in the case of diabetic retinopathy is considered as a key method for the treatment of this disease. Additionally, approaches are known that enhance the functioning of MGCs by stimulating the beta-adrenergic pathway [276].

To date, along with the already studied systemic and regulatory factors of the MGCs (CNTF, GDNF, FGF2, IGF, Notch-Delta, Wnt/b-catenin, etc.) [7,9,51,277], new ones such as miRNAs have also become known [278,279]. Overexpression of miR-25 (or let-7) in antagonism with miR-124 induces the expression of the proneural TF Ascl1, providing reprogramming of mature MGCs into NR cells in mice [280]. miR-9/9* and miR-124 act as negative regulators of the Sox2-Ascl1a/Atoh7-Lin-28 pathway, inhibiting proliferation, and as activators of the TLX-ONECUT signaling, stimulating MGC differentiation into retinal neurons [281]. In addition to their role in the regulation of MGC behavior, miRNAs are involved in the progression of glaucoma and ocular pressure regulation. In the very-near future, a study of this issue and, in particular, the delivery of miRNAs in order to develop miRNA-based glaucoma therapies is required [282]. 

Epigenetic features of MGCs are also subject to consideration. Many authors indicate the similarity in molecular genetics between mature MGCs and late progenitors for bipolars and photoreceptors (rods) [259,260,283,284]. Data by Dvoriantchikova et al. [284], characterizing the epigenetic profile of MGCs, confirm this. The results of ChIP-seq suggest that the obstacles to NR regeneration by MGCs in mammals are explained by the state of chromatin suppressing the work of genes responsible for the late stages of MGC reprogramming, in particular, the specification of retinal neuron types. This problem may be addressed by means of pioneering TFs and DNA demethylase activity [284]. There is evidence of changes in the DNA methylation/demethylation [285]. MGCs are assumed to maintain the plasticity of differentiation that allows them to be reprogrammed and, as a result, regenerate dead NR cells by inducing additional epigenetic modifications of the genome [285,286]. Jorstad et al. [286], using a histone deacetylase inhibitor and providing overexpression of the key regulator of MGC reprogramming, *Ascl1* gene [287], managed to stimulate neurogenesis in the adult mouse MGC population destroyed by a neurotoxin at the stage of their development, when the natural manifestation of neurogenic potential had already been blocked. The descendant cells of MGCs, pretreated this way, expressed the marker proteins of interneurons, formed synapses with pre-existing NR neurons, and responded to a light stimulus [286].

The results of a comparative study of the genetic network controlling the regenerative activity of MGCs in fish, birds, and mice have been published recently [288]. The differences in gene expression and chromatin availability in MGCs in response to various stimuli in these animals were assessed by RNA-seq and ATAC-seq. Evolutionarily conserved genetic networks with specific features and that control MGCs were identified in a resting state, in a reactive state, and during neurogenesis. In all animals, NR damage induces reactivation and reprogramming of MGCs; however, in the mouse regulation network, including nuclear factors 1 (NF1), cells return to a resting state. The deletion of the NF1-encoding genes allows mouse MGCs to complete the final stages of cell conversion, i.e., produce retinal neurons. Thus, NF1 is another key factor in the regulation network involved in suppression of neurogenesis in the mammalian MGC population [288].

An unusual study by Bonilla-Pons et al. [289] has been conducted on an organotypic culture of NR using cell fusion technology. As a result, it turns out that, as in mice [290], human MGCs can fuse with human mesenchymal stem cells, giving rise to hybrid cells. Such hybrids are capable of producing retinal neurons, and their pretreatment with the activator Wnt/β-catenin increases their differentiation into neuronal-like cells, providing the ability of the hybrid descendants to migrate, engraft, and manifest neural, proto-electrophysiological properties. The authors [289] suggest that this technique of cell fusion-mediated MGC reprogramming and neurogenesis in the human NR is another potential therapeutic approach to cell regeneration for the treatment of human NR diseases. It should also be mentioned that the presence of stem cells is known to enrich the cell environment in trophic molecules and self-renewal factors and to enhance cell homing and survival [291,292]. In turn, the enriched environment exhibits neuroprotective properties, as has been evidenced, in particular, by an induced model of ocular hypertension [293].

The new information that human surgical NR explants having ciliary epithelium in their structure contain cells with the retinal precursor cell properties seems encouraging [88]. These cells from surgical explants obtained through vitrectomy for the management of rhegmatogenous detachment, when placed under cultivation conditions without growth factors, are able to migrate, proliferate, and express progenitor markers: Pax6, Sox2, CRX, nestin, p27^kip1^, and also marker proteins of MGCs (GFAP, GS). The authors note that these cells obtained from patients can be a source of MGCs and other retinal progenitors [88].

## 5. Conclusions

The results of in vivo and in vitro studies indicate that intrinsic retinal regeneration cell sources (RRCSs), which include RPE, CB, MG, and the NR ciliary region, have intrinsic genetic features that determine their potencies for retinal neuron production. These potencies are implemented, to varying degrees, from complete or partial retinal regeneration by RRCSs in fish, amphibians, and bird embryos to the manifestation of certain progenitor properties, proliferation, and change of the RRCS phenotype in mammals. In the latter, as well as in humans, such regenerative responses are most frequently found under conditions of directed induction in vitro. Identification of RRCSs and mechanisms of regulation of their behavior have long been conducted in various studies on animal models and humans. The studies have shown that the mechanisms regulating the manifestation/inhibition of regenerative responses of RRCSs in animals and humans are common. In general, these include external signaling, changing transcription patterns, and the epigenetic landscape. The pathways of conversion through proliferation and acquisition of a progenitor state that cells can leave by acquiring a new differentiation of one or more retinal phenotypes are also common for latent RRCSs (RPE, CB, and MGCs). In the progenitor state phase, RRCSs express “developmental” TFs and multipotency genes that are controlled by intracellular mechanisms of genome regulation. These mechanisms, in turn, depend on a wide range of external effects such as signaling molecules (growth factors, inflammation, viability, cell death, hormones, etc.). Modulation of immune responses during degenerative processes in the NR can also affect the course of NR regeneration. 

The major degenerative diseases of the retina are associated with changes, death, and loss of function by the RPE and photoreceptors. Diseases of this kind include AMD, PVR, and RP. Glaucoma is caused by the loss of ganglion cells, while reactive gliosis, accompanying many NR pathologies, is caused by cell hypertrophy and an increase in the MGC population. To translate data to biomedicine of the eye, studies of RRCSs are being conducted in two main areas. The first is a search for technologies to provide replacement of dead/degenerating NR cells with healthy ones obtained from endogenous RRCSs. Designing the methods to promote the production of autologous retinal cells de novo under in vivo or in vitro conditions is an important alternative to the use of stem cells for these purposes. The use of exogenous stem cells in ophthalmology is widely studied currently. Nevertheless, despite marked technological advances, the risks of mutations, tumor growth, undesirable benign transformations of these cells, etc. have been recorded. Obtaining the required cell types from stem cells with integration of new cells into the cell ensemble and, at last, the issue of immune rejection, still pose serious challenges. The natural retinal regeneration process achieved through the use of RRCSs may be more successful for therapy, including the facilitated integration of endogenous cells and proper synaptic targeting. However, the issue of competition between the use of endogenous RRCSs and stem cells is not addressed here. As can be seen from the examples discussed above, these methods can be coupled and complement each other.

There is another important implication of the knowledge of RRCSs in biomedicine of the eye. The information about the molecular-genetic status of RRCSs and the molecular mechanisms regulating their behavior makes it possible to prevent RRCS-associated pathological processes, such as AMD, PVR, reactive gliosis, and glaucoma. In the case of such pathologies, the knowledge of the RRCS control mechanisms contributes to the development of therapy targeted at inhibiting the active manifestation of RRCS-related disorders. Despite the evident successes in the study of the mechanisms of internal and external regulation of RRCS behavior, further research in this field is still required to elucidate, in particular, the fate of RRCS-derived cells during transplantation and their integration in the damaged mammalian retina. It is obvious that the direct use of RRCSs is not a close prospect today. However, as discussed in the review, with the application of techniques for intracellular and external regulation of RRCS behavior in animal models, the desired goals of cell replacement and/or correction of degenerative changes in the NR can be achieved.

## Figures and Tables

**Figure 1 cells-11-03755-f001:**
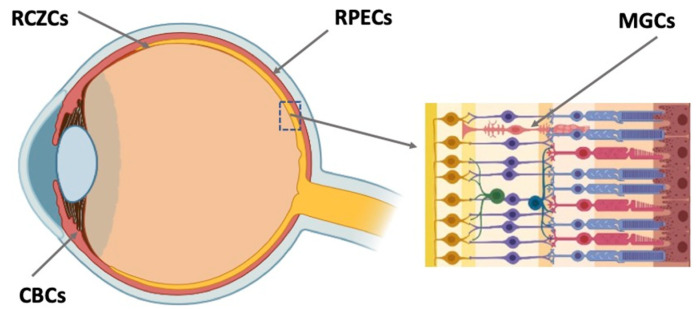
Potential endogenous cell sources for retinal regeneration (summarized data). RCZCs—cells of the retina ciliary zone; RPECs—retinal pigment epithelial cells; CBCs—cells of the ciliary body; MGCs—Müller glial cells.

**Figure 2 cells-11-03755-f002:**
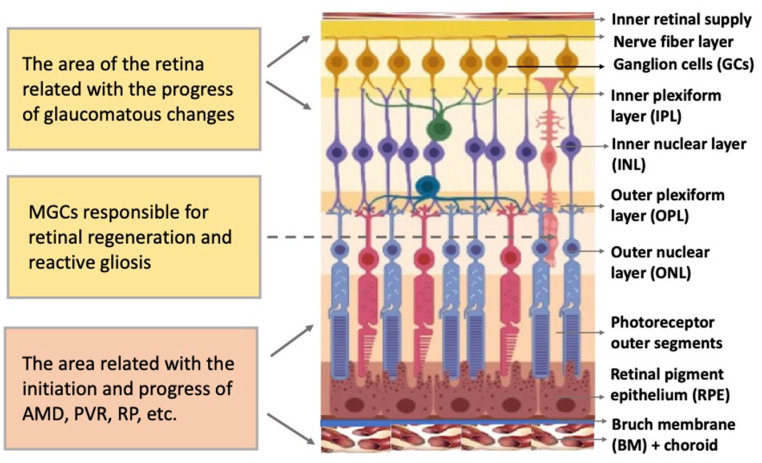
Structure of the retina and retinal compartments involved in degenerative diseases. MGCs—Müller glial cells, AMD—age-related macular degeneration, PVR—proliferative vitreoretinopathy, RP—retinitis pigmentosa.

**Figure 3 cells-11-03755-f003:**
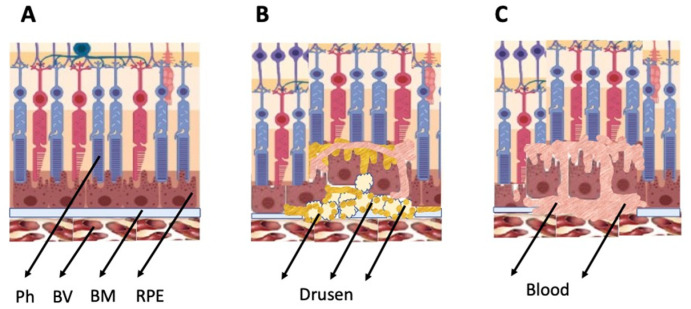
Schematic representation of AMD-related changes in the eye. (**A**)—normal eye; (**B**)—“dry” AMD; (**C**)—“wet” AMD. Ph—photoreceptors, BV—blood vessels, BM—Bruch membrane; RPE—retinal pigment epithelium.

**Figure 4 cells-11-03755-f004:**
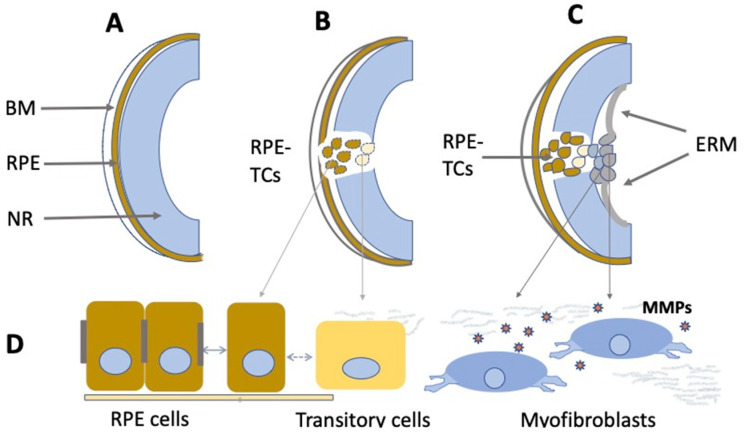
Schematic illustration of PVR development. (**A**)—Normal eye. BM—Bruch’s membrane, RPE—retinal pigment epithelium, NR—neural retina. (**B**)—RPE at the beginning of the epithelial– mesenchymal transition. RPE-TCs—RPE-derived transforming cells. (**C**)—Epiretinal membrane (ERM) formation. (**D**)—Morphological changes of RPE cells during the epithelial–mesenchymal transition. MMPs—Matrix metalloproteases.

**Figure 5 cells-11-03755-f005:**
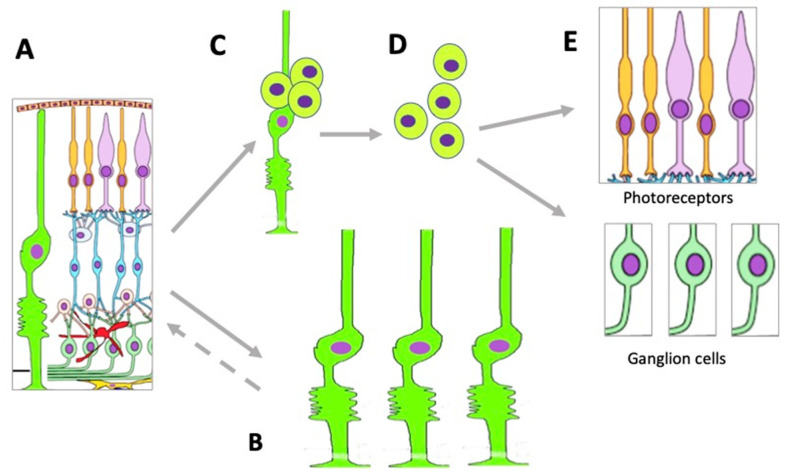
Changes occurring in the MGC population under conditions of retinal damage. (**A**)—MGCs in the structure of normal retina; (**B**)—MGC hypertrophy and proliferation in conditions of reactive gliosis; (**C**)—MGC reprogramming and proliferation during retinal regeneration in vivo and after directed stimulation in vitro; (**D**)—MGC-derived retinal cell precursors emerging during retinal regeneration in vivo and after directed stimulation in vitro; (**E**)—retinal neurons formed from MGC-derived retinal cell precursors. See more detailed description in the text.

## Data Availability

Not applicable.

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
