# Peer review of "Cell Sources for Retinal Regeneration: Implication for Data Translation in Biomedicine of the Eye"

_cells, 2022, doi:10.3390/cells11233755_

Round 1

Reviewer 1 Report

This is a well written review article that discuss an interesting topic that can bring attention of future researchers for developing new therapies for retinal degeneration diseases

Author Response

I express my great gratitude to the reviewer for the high valuation of the manuscript. Since there were no questions or comments when reviewing it (they are not reflected on the corresponding page of the Cells website), all the edits I made meet the recommendations of two other reviewers. Minor grammatical errors and typos have been corrected through the text.

Reviewer 2 Report

According to the authors statement, the present review provides information about intrinsic RRCSs and the feasibility of using this information in biomedicine.

The review is very poor not in its length but in its content. It mostly reflects a sort of academic lesson without adding any value to our present knowledge. The fact that it is not based on original papers by the same author or her group renders the review a pure writing exercise that cannot be accepted by the scientific community. Even in the case of some sort of summary of what is known about the argument, much information are lacking and the view of the argument needs more work.

In detail:

1.The description of the retinal organization is too extensive but it completely lacks of the organization of vessels that supply retinal cells and participate to the formation of the neurovascular unit that is impaired in retinal diseases.  Major degenerative diseases include damage to retinal vascularization, an issue that is completely ignored in the present review

2. Why retinal diseases are “caused  by the loss of cells and cell–cell interactions in the functional light perception system” with the exception of glaucoma?

3. Abbreviations need to be reformulated according to their extended definition if one. There are several examples of misleading abbreviatios. For example, which is the meaning of NR neurons ? neural retina neurons? Abbreviations should be in line with which are used in the literature

4. Cell sources for retinal cell regeneration listed in paragraph 1.3 mostly reflect a copy and paste from previous works. Data reviewed here do not reflect original works by the author. Beside her poor production, papers from the author include mostly reviews.

5. No take-home message from this review that does not increase our knowledge but rather confuses it

Author Response

Reviewer 2.

First of all, I am very grateful to the reviewer for an attentive attitude, great criticism and certain, critical point of view regarding my manuscript. Undoubtedly, I take into account all the points of the review. My answers are briefly given below.

The original works of my team and me is a result of two major directions in the study of retinal regeneration: 1) retinal regeneration in lower vertebrates, 2) reprogramming of mammalian (including human) RPE cells in vivo and in vitro. Results of these studies were published, cited earlier, and included in the main context of the present manuscript. During our work we published more than 150 papers in International and Russian scientific, academic journals. For this reason, I would like to hope that all of them are not only "a pure writing exercise", but have meaningful value. The subject of this manuscript is not someone else's or copied.

In detail:

1.The description of the retinal organization is too extensive but it completely lacks of the organization of vessels that supply retinal cells and participate to the formation of the neurovascular unit that is impaired in retinal diseases.  Major degenerative diseases include damage to retinal vascularization, an issue that is completely ignored in the present review.

In section 1.1. information about the vascular system supplying the retina is somewhat expanded (lines:86-90 and fig. 2). Indeed, the issue of the consequences of damage to retinal vascularization in the development of degenerative retinal diseases is not considered in the paper. The emphasis in the manuscript is made on the issue of cell replacement therapy, however there are other important issues beyond. It is not only the question of necessary support from the circulatory system of the retina, but the establishment of neuronal connections and transmission of correct information to the visual analyzer also. These and other emerging problems are very big, significant, and require a consideration in special reviews.

  1. Why retinal diseases are “caused by the loss of cells and cell–cell interactions in the functional light perception system” with the exception of glaucoma?

When using the term "functional light perception system", I meant the system of functional relationships between photoreceptors and retinal pigment epithelium, i.e. in the part of the retina that is called the outer retina. Glaucoma, in its severe stage, leads to the retrograde death of the ganglion cells and then a part of the interneurons. These cells represent the inner retina – a cascade of cells and their functional interactions responsible for the transmission of the visual signal.

  1. Abbreviations need to be reformulated according to their extended definition if one. There are several examples of misleading abbreviations. For example, which is the meaning of NR neurons? neural retina neurons? Abbreviations should be in line with which are used in the literature.

Abbreviations have been clarified and, if it was possible, brought into a line with the most widely used in the literature. Many abbreviations were deleted as superfluous and cluttering up the text. All corrections are marked in the margins of the paper. The phrase “NR neurons" is definitely nonsense, it was replaced by “NR cell populations". The abbreviation "NR" – neural retina, instead of the "retina", is used quite widely in a literature and is necessary for distinguishing the retinal pigment epithelium and the neural part of the retina as tissues of different specialization and phenotypes.

  1. Cell sources for retinal cell regeneration listed in paragraph 1.3 mostly reflect a copy and paste from previous works. Data reviewed here do not reflect original works by the author. Beside her poor production, papers from the author include mostly reviews.

Paragraph 1.3 is the semantic part of the manuscript, consisting of subsections. They contain summarized, but not simply copied information. The later is taken in balance from the original articles and reviews. The original works of my laboratory are presented in a condensed form in section 1.3.3. and cited under the numbers 29, 66-68, 133, 158, 178, 225, 228-230. I would also like to note that journal rules for authors contain a warning against “excessive citation by the author of his own works and/or his team”.

Minor grammatical errors and typos have been corrected through the text. 

  1. No take-home message from this review that does not increase our knowledge but rather confuses it.

This critical remark reflects the reviewer's personal impressions and assessments. For this reason, I can only take into account this point of view, add it to the overall picture of external impressions of my work, and to express my sincere gratitude once again.

Reviewer 3 Report

Overview: In this review article, the authors have discussed about the potential sources of regenerative stem cells (embryonic, neural, mesenchymal and induced) for the replacement therapy of retinal degenerative diseases. This is a nicely written narrative review article and I learned something new in the review process. However, there are some minor corrections needed. Please see the specific comments and address those issues.  

Specific comments:

Abstract:

L6: I wonder if we can include retinal detachment (RD) in the degenerative list! The RD can be caused by varied etiopathogenesis. For example, trauma related RD is not associated with degeneration although it may lead to degeneration later if left untreated.

L7-8: “ Novel approaches to  the treatment of the retinal diseases are currently developed based on cell replacement therapy.” So far the cell replacement therapy has not come to the clinics effectively. They are still being tested. So you need to modify this sentence into:

“Novel approaches to the treatment of the retinal diseases are currently being developed based on cell replacement therapy.”

L10: The abbreviation RRCSs is difficult to follow throughout the text because the abbreviated letters are not in the order of the words.

Introduction:

L41: I would prefer to write Muller cells or Muller glial cells instead of Muller glia cells. Similarly, throughout the manuscript where you have written “glia”, including Figure 1.

L55-56: I think you have not defined SC (Stem Cells). Please define the abbreviation in its first mention in L55. 

L56: It is not defined what is ‘P’ in induce d (iPSCs).

L93: Again, I am not comfortable to classify retinal detachment as primary degenerative disease.

L97-98: “There are two forms of AMD: the so-called ‘dry’, prevailing, form and ‘wet’ AMD [25]”.  Please remove loose writing “so-called”. Secondly, what do you mean by prevailing, form?

L99: Define ECM.

L124/126: What is PR? Do you mean RP?

L241: What is EGFR?

L771-772: This is a better word arrangement for RRCSs. This should come in the first mention to avoid confusion.

It may be a more balanced review article if the authors talk about the limitations of each progenitor or stem cell types.

Author Response

Reviewer 3.

I express my deep gratitude to the reviewer for the hard work of reading the review and for all the comments and suggestions made for necessary corrections. Below – the list of all questions and my answers for each of them.

Overview: In this review article, the authors have discussed about the potential sources of regenerative stem cells (embryonic, neural, mesenchymal and induced) for the replacement therapy of retinal degenerative diseases. This is a nicely written narrative review article and I learned something new in the review process. However, there are some minor corrections needed. Please see the specific comments and address those issues.  

Specific comments:

Abstract:

L6: I wonder if we can include retinal detachment (RD) in the degenerative list! The RD can be caused by varied etiopathogenesis. For example, trauma related RD is not associated with degeneration although it may lead to degeneration later if left untreated.

I understand the disagree of the reviewer with the listing retinal detachment among other degenerative retinal diseases. When I did that, there was only an association of RD with proliferative vitreoretinopathy. I removed the RD from the list of cellular causes of retinal degenerations. It is marked in the margin.

L7-8: “Novel approaches to the treatment of the retinal diseases are currently developed based on cell replacement therapy.” So far, the cell replacement therapy has not come to the clinics effectively. They are still being tested. So, you need to modify this sentence into: 

“Novel approaches to the treatment of the retinal diseases are currently being developed based on cell replacement therapy.”

I agree. This sentence is modified. Thank you.

L10: The abbreviation RRCSs is difficult to follow throughout the text because the abbreviated letters are not in the order of the words. 

The sentence is reformulated to bring the abbreviation in accordance with the word order: “An alternative and complement to this approach can be the use of the intrinsic reserve, retina regeneration cell sources (RRCSs) such as…”

Introduction:

L41: I would prefer to write Muller cells or Muller glial cells instead of Muller glia cells. Similarly, throughout the manuscript where you have written “glia”, including Figure 1.

The name of Müller cell population has been specified as Müller glial cells or MGCs and unified in words and abbreviations throughout the text and figures.

L55-56: I think you have not defined SC (Stem Cells). Please define the abbreviation in its first mention in L55.  L56: It is not defined what is ‘P’ in induce d (iPSCs). 

This sentence (55-56 lines) is rewritten, abbreviations of various types of stem cells are deleted as superfluous and cluttering up the text. The world “pluripotent” – added. Superfluous abbreviations of some molecules are deleted also.

L93: Again, I am not comfortable to classify retinal detachment as primary degenerative disease.

I agree and to be more accurate, I deleted “retinal detachment” from the list of retinal degenerative diseases.

L97-98: “There are two forms of AMD: the so-called ‘dry’, prevailing, form and ‘wet’ AMD [25]”.  Please remove loose writing “so-called”. Secondly, what do you mean by prevailing, form? 

“So-called” is deleted. The word “prevailing” I used as predominant form of AMD. The information is taken from the cited review.

L99: Define ECM.  Abbreviation ECM – defined.

L124/126: What is PR? Do you mean RP? Yes, it is. It is just a misspell, it is corrected

L241: What is EGFR? Abbreviation was disclosed.

L771-772: This is a better word arrangement for RRCSs. This should come in the first mention to avoid confusion. 

Sure, and I’ve done so (see abstract and further in the text), thank you!

It may be a more balanced review article if the authors talk about the limitations of each progenitor or stem cell types.

Undoubtedly, limitations and measure of complexity/risk in the practical use of the particular progenitor or stem cell type is the most important question. However, a consideration of this question is quite difficult in the current period of time due to insufficient accumulation of both practical/experimental, and fundamental results. A deep analysis of this problem is the good prospect for separate special reviews in future.Minor grammatical errors and typos have been corrected through the text.

Round 2

Reviewer 2 Report

No major changes to the present version

Author Response

I thank the reviewer again. The review significantly helped to improve the text of the article. As I can understand there are no more additional questions or suggestions in the second round.